# Maximum Entropy Information Bottleneck for Confidence-aware Stochastic Embedding

## Abstract

Stochastic embedding has several advantages over deterministic embedding, such as the capability of associating uncertainty with the resulting embedding and robustness to noisy data. This is especially useful when the input data has ambiguity (e.g., blurriness or corruption) which often happens with in-the-wild settings. Many existing methods for stochastic embedding are limited by the assumption that the embedding follows a standard normal distribution under the variational information bottleneck principle. We present a different variational approach to stochastic embedding in which maximum entropy acts as the bottleneck, which we call "Maximum Entropy Information Bottleneck" or MEIB. We show that models trained with the MEIB objective outperform existing methods in terms of regularization, perturbation robustness, probabilistic contrastive learning, and risk-controlled recognition performance.

## 1 Introduction

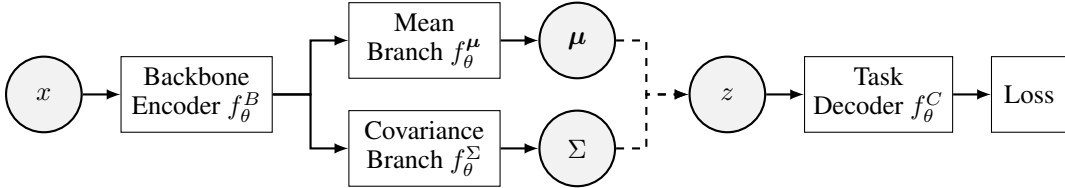

Figure 1: Stochastic embedding framework.

Stochastic embedding is a mapping of an input $x$ to a *random variable $Z \sim p(z|x) \in R^D$* in which the mapped *regions* of similar inputs are placed nearby. Unlike deterministic embedding, where $z = f(x)$ is a point in $R^D$, stochastic embedding can represent the input uncertainty, such as data corruption or ambiguity, by controlling the spread of probability density over a manifold Oh et al. (2019).

Figure 1 depicts a typical stochastic embedding framework with the neural networks parameterized by $\theta$. Input $x$ is mapped to a Gaussian distribution $\mathcal{N}(z; \boldsymbol{\mu}, \Sigma)$ by a stochastic encoder that consists of a backbone feature extractor $f_\theta^B$ followed by two separate branches $f_\theta^{\boldsymbol{\mu}}$ and $f_\theta^\Sigma$, each of which predicts the $\boldsymbol{\mu}$ and $\Sigma$.[1] While the covariance matrix $\Sigma$, in prior work as well as in this paper, is assumed to be diagonal where $f_\theta^\Sigma$ outputs a $D$-dimensional vector, it would be straightforward to extend it to a full covariance matrix, for instance, using a Cholesky decomposition Dorta et al. (2018). Embeddings sampled from this Gaussian are then consumed by a decoder $f_\theta^C$ for the downstream task, e.g., classification.

Majority of leading methods for stochastic embedding Oh et al. (2019); Chang et al. (2020); Sun et al. (2020); Chun et al. (2021); Li et al. (2021b) are built upon the variational information bottleneck (VIB) principle Alemi et al. (2017) where the stochastic encoder $p(z|x)$ is regularized by Kullback–Leibler (KL) divergence, $\text{KL}(p(z|x)||r(z))$, where $p(z|x) = N(z; \boldsymbol{\mu}, \Sigma)$ and $r(z) = N(z; \mathbf{0}, \mathbf{I})$ in general. This effectively impels the embeddings to be close to a standard normal distribution, which is an explicit assumption that may not always hold true.

---

[1] We use the terms $f_\theta^{\boldsymbol{\mu}}(x)$ and $f_\theta^\Sigma(x)$ interchangeably with $f_\theta^{\boldsymbol{\mu}}(f_\theta^B(x))$ and $f_\theta^\Sigma(f_\theta^B(x))$ respectively.

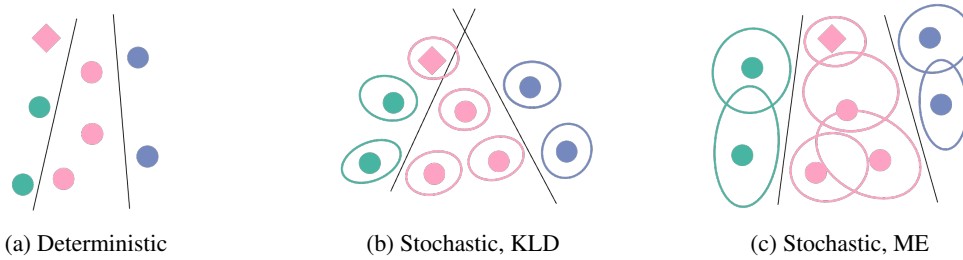

(a) Deterministic  (b) Stochastic, KLD  (c) Stochastic, ME

Figure 2: Embedding space characteristics. Each color represents a class of data. The color-filled shapes refer to the deterministic or the mean point of stochastic embeddings. The ellipses around the shapes depict the standard deviation of stochastic embeddings. The circles and the diamonds represent training and testing data, respectively. The solid lines are the decision boundaries learned.

Furthermore, Bütepage et al. (2021) showed that the standard variational autoencoder (VAE) trained with KL divergence from the standard normal prior Kingma & Welling (2014) fails to correlate the latent variance with the input uncertainty; the variance decreases with the distance to the latent means of training data, which is contrary to expectation. Since VAE is a special case of an unsupervised variant of VIB, this phenomenon also holds for VIB; our experiments show VIB assigns smaller variance to more uncertain inputs (see the supplemental Section A). Motivated by this finding, we explicitly use the variance (entropy) as a confidence indicator rather than a measure of input uncertainties and encourage the model to assign larger variance to more certain inputs.

In this paper, we propose *Maximum Entropy Information Bottleneck (MEIB)* to lift such constraints of using a fixed prior and instead use the conditional entropy of the embedding $H(Z|X)$ as the only regularization. Based on the maximum entropy principle Jaynes (1957), we postulate that stochastic uncertainty is best represented by the probability distribution with the largest entropy. By maximizing $H(Z|X)$, the embedding distribution is promoted to be more random, pushing for broader coverage in the embedding space, with a trade-off on the expressiveness of Z about target Y. The resulting distribution is also the one that makes the fewest assumptions about the true distribution of data Shore & Johnson (1980).

Figure 2 depicts our intuition; (a) deterministic encoders would learn embeddings "just enough" to classify the training samples unless any regularization technique, such as a margin loss, is considered. It would be vulnerable to small changes in test inputs. (b) the embedding distribution by typical stochastic encoders (e.g., VIB) trained with the KL divergence regularization will tend to cover a fixed prior. Note that it is generally difficult to pick a true prior distribution. Also, it is unnecessary to restrict the embedding distribution to be within a specific bound. (c) with MEIB, on the other hand, by maximizing the conditional entropy of the stochastic embeddings, we would have a better regularization effect as it makes the area *secured* by the embedding distribution for the given input as broad as possible.

The key contributions of MEIB to the previous stochastic embedding methods are summarized as follows:

- While it provides a comparable regularization in handwritten digit classification, MEIB outperforms existing approaches in the challenging person re-identification task with three popular datasets.

- MEIB shows significantly better perturbation robustness compared to VIB in handwritten digit classification.

- MEIB performs better than VIB when used in a probabilistic contrastive learning framework.

- Providing reliable confidence measurements, MEIB shows an outstanding risk-controlled recognition performance in digit classification and person re-identification tasks.

## 2 RELATED WORK

**Stochastic Embeddings**   Research on stochastic embeddings has gained popularity in recent years. Oh et al. (2019) proposed a similarity-based method based on the pairwise soft contrastive loss and the VIB principle, which has been later applied to human pose embedding Sun et al. (2020), cross-modal retrieval Chun et al. (2021), and ordinal embedding Li et al. (2021b). Chang et al. (2020) proposed stochastic embedding for face recognition using softmax loss and KL divergence regularization without relying on similarity. All of these methods assume embedding distribution to be unit Gaussian. Unlike these approaches, Probabilistic Face Embedding (PFE) Shi & Jain (2019) turns deterministic embeddings into Gaussians with fixed mean by training a post hoc network to maximize the mutual likelihood of same-class embeddings. Follow-up research on PFE includes its extension to triplets Warburg et al. (2021) and for spherical space Li et al. (2021a). However, this line of work is limited to fixed embedding mean. The work most closely related to ours is DistributionNet Yu et al. (2019) which introduced entropy-based regularization and inspired Yang et al. (2021) for their uncertainty-aware loss. However, these methods put a margin to bound the total entropy rather than maximizing it as we do.

**Maximum Entropy**   Maximum entropy is a general principle that has already been widely adopted in designing machine learning models, including supervised and reinforcement learning Zheng et al. (2017); Ahmed et al. (2019). Pereyra et al. (2017) used the negative entropy of the class prediction distribution, $-H(p_\theta(y|x))$, as a regularization term in the loss function to prevent over-confident predictions. In reinforcement learning, the maximum entropy framework encourages diverse explorations in both on-policy and off-policy settings Mei et al. (2019); Haarnoja et al. (2018); Han & Sung (2021). However, in most previous work, the entropy regularization has been applied at the decision levels, the distribution of class or action predictions. In this work, on the other hand, we focus on the entropy of the stochastic embedding of inputs.

## 3 MAXIMUM ENTROPY INFORMATION BOTTLENECK

**MEIB Objective**   *The maximum entropy principle* Jaynes (1957) states that the current state of knowledge about the given system is best represented by the probability distribution with the largest entropy Shore & Johnson (1980). By combining this with our hypothesis, the goal is to learn an encoding $Z$ that is maximally expressive about $Y$ while maximizing the expected amount of information (entropy) about $Z|X$:

$$\max_\theta I(Z, Y; \theta) \quad \text{s.t.} \quad H(Z|X; \theta) \geq H_c. \tag{1}$$

Introducing a Lagrangian multiplier $\beta$, we have the maximization objective:

$$\mathcal{J}_{\text{MEIB}} = I(Z, Y; \theta) + \beta H(Z|X; \theta) \tag{2}$$

where $\beta \geq 0$ controls the trade-off between the predictiveness and the spread of $Z$ given $X$. Using the lower bound suggested by Alemi et al. (2017) for the first term of the objective $I(Z, Y)$, we have

$$I(Z, Y; \theta) + \beta H(Z|X; \theta) \geq \int dx\, dy\, dz\, p(x)p(y|x)p(z|x) \log q(y|z)$$
$$- \beta \int dx\, dz\, p(x)\, p(z|x) \log p(z|x) = L. \tag{3}$$

This lower bound $L$ can be computed by approximating the joint distribution $p(x, y) = p(x)\, p(y|x)$ using the empirical data distribution $p(x, y) = \frac{1}{N} \sum_{n=1}^{N} \delta_{x_n}(x)\, \delta_{y_n}(y)$ (Alemi et al., 2017)

$$L \approx \frac{1}{N} \sum_{n=1}^{N} \left[ \int dz\, p(z|x_n) \log q(y_n|z) - \beta\, p(z|x_n) \log p(z|x_n) \right]. \tag{4}$$

Consequently, the loss function to be minimized is:

$$\mathcal{L}_{\text{MEIB}} = \frac{1}{N} \sum_{n=1}^{N} \left[ \mathbb{E}_{z \sim p(z|x_n)} \left[ -\log q(y_n|z) \right] - \beta H(Z|x_n) \right] \tag{5}$$

where we use the typical reparameterization trick Kingma & Welling (2014) to backpropagate gradients through the sampling of $z \sim p(z|x_n)$. We use a single sample of $z$ by default unless it is specified.

**Relationship to VIB**   The minimization loss function of VIB (Alemi et al., 2017) is:

$$\mathcal{L}_{\text{VIB}} = \frac{1}{N} \sum_{n=1}^{N} \left[ \mathbb{E}_{z \sim p(z|x_n)} \left[ -\log q(y_n|z) \right] + \beta \text{KL}[p(Z|x_n), r(Z)] \right] \tag{6}$$

$$= \frac{1}{N} \sum_{n=1}^{N} \left[ \mathbb{E}_{z \sim p(z|x_n)} \left[ -\log q(y_n|z) \right] - \beta H(Z|x_n) + \beta H(p(Z|x_n), r(Z)) \right] \tag{7}$$

$$\geq \frac{1}{N} \sum_{n=1}^{N} \left[ \mathbb{E}_{z \sim p(z|x_n)} \left[ -\log q(y_n|z) \right] - \beta H(Z|x_n) \right] = \mathcal{L}_{\text{MEIB}} \tag{8}$$

where $H(p(Z|x_n), r(Z))$ is the cross-entropy of $r(Z)$ relative to the distribution $p(Z|x_n) = \mathcal{N}(Z|f_\theta^{\boldsymbol{\mu}}(x_n), f_\theta^{\Sigma}(x_n))$, which is given by

$$H(p(Z|x_n), r(Z)) = \frac{1}{2} \left( D \ln(2\pi) + \sum_{d=1}^{D} \left( \boldsymbol{\mu}_{\theta,d}^2 + \boldsymbol{\sigma}_{\theta,d}^2 \right) \right) \geq 0$$

where $\boldsymbol{\mu}_\theta = f_\theta^{\boldsymbol{\mu}}(x_n)$ and $\Sigma_\theta = \text{Diag}(\boldsymbol{\sigma}_\theta^2) = f_\theta^{\Sigma}(x_n)$, a diagonal covariance, respectively - a detailed derivation can be found in the supplemental material Section B. Therefore, the VIB loss function is an upper bound of the MEIB loss function with a positive value of $\beta$.

**Confidence Measure of MEIB**   MEIB encourages obvious inputs that can be easily classified to take broader embedding areas by assigning larger entropy. On the contrary, the inputs closer to other classes would have smaller entropy to reduce the chance of misclassification according to the loss function. Therefore, we adopt the conditional entropy $H(Z|x)$ as our confidence measure for the input $x$, which is given by Cover & Thomas (2006):

$$H(Z|x) = \frac{1}{2} \ln(2\pi e)^D |\Sigma_\theta| = \frac{1}{2} |\Sigma_\theta| + \frac{D}{2}(1 + \ln 2\pi). \tag{9}$$

Specifically, we use the *dimension-wise average* conditional entropy $H(Z|x)/D$ to achieve a dimension-agnostic confidence measure.

## 4   EXPERIMENTAL RESULTS AND DISCUSSION

In this section, we show experimental results for various tasks to demonstrate the effectiveness of MEIB in terms of the regularization, perturbation robustness, and confidence measure. All computational experiments were implemented using PyTorch Paszke et al. (2019) 1.9 with Python 3.7 on a workstation equipped with an NVIDIA® GeForce® RTX 2080 Ti graphic card. Please refer to the supplemental material Section H for the implementation and training details for each experimental task.

### 4.1   DIGIT CLASSIFICATION

First, we evaluate MEIB on a handwritten digit classification as the simplest form of benchmark task. We use the QMNIST dataset Yadav & Bottou (2019) to utilize its larger set of 60,000 testing data compared to 10,000 of those in the original MNIST dataset, while it has almost identical training data. We adopt the same architecture employed in Alemi et al. (2017); the backbone encoder $f_\theta^B$ is a multilayer perceptron (MLP) with two fully-connected (FC) layers of 1024 hidden units with ReLU activations. Both $f_\theta^{\boldsymbol{\mu}}$ and $f_\theta^{\Sigma}$ are an FC layer of $D$ hidden units where the exponential (exp) function was applied after $f_\theta^{\Sigma}$. We found that applying a batch normalization (BN) layer at the end of $f_\theta^{\boldsymbol{\mu}}$ improves the performance of MEIB with a noticeable gap (see the supplementary material Section F). The decoder $f_\theta^C$ is an FC layer with the softmax function that outputs $p(y|x)$ over ten classes.

Table 1: QMNIST test set error rate (%)

| Method | $D = 2$ | $D = 256$ |
|---|---|---|
| Deterministic | $4.64 \pm 0.43$ | $1.71 \pm 0.04$ |
| Dropout | $4.08 \pm 0.29$ | $1.62 \pm 0.02$ |
| VIB | $3.29 \pm 0.32$ | $1.75 \pm 0.03$ |
| MEIB | $3.95 \pm 0.41$ | $1.76 \pm 0.05$ |
| VIB (12 MC samples) | $3.21 \pm 0.32$ | $1.45 \pm 0.02$ |
| MEIB (12 MC samples) | $3.31 \pm 0.35$ | $1.48 \pm 0.04$ |

Using the same architecture, we compare the following embedding approaches with MEIB: a deterministic baseline, dropout Srivastava et al. (2014), and VIB. The deterministic baseline represents the typical usage of neural network models without stochasticity. By omitting the variance estimator module $f_\theta^\Sigma(x)$, the embedding of input is deterministically given by $z = f_\theta^{\boldsymbol{\mu}}(x)$. Dropout is one of the most popular regularization methods for neural networks; thus, it is considered the first benchmark regularization method to compare Alemi et al. (2017); Ghiasi et al. (2018). We use the same deterministic model, but dropout is applied with the probability of 0.5 during the training time. Unlike MEIB and VIB, both deterministic models were trained only with the cross-entropy loss. We set $\beta = \alpha/D$ where $\alpha$ is equal to 0.01 for VIB, 0.1 for MEIB with $D = 2$, and 1 for MEIB with $D = 256$. Please refer to the supplementary material Section F for the hyperparameter study. All models were trained with five different random seeds, and we report the mean and standard deviation for each performance metric.

**Regularization Effect** Table 1 shows the classification results by each method on the QMNIST test set. Specifically, we compared the methods with two different embedding sizes, $D = 2$ and 256. With $D = 2$, the stochastic methods performed better than the deterministic ones regardless of the usage of dropout. Using the larger embedding size of $D = 256$, on the other hand, the deterministic model trained with dropout performed the best while the others have very similar performance considering the standard deviation. However, using 12 Monte Carlo (MC) samples of $z$, the stochastic methods outperformed the deterministic ones with a larger gap than the single sample case. MEIB provides a reasonable amount of regularization comparable to VIB.

**Perturbation Robustness** The test time noise/perturbation robustness of a deep neural network model is an important aspect, especially when the model is considered to be deployed for a real-world application. We evaluate the robustness of models toward adversarial examples as an alternative form of perturbation robustness evaluation because models that are weak to adversarial examples might be vulnerable to not only intended attacks but also unexpected noise in test time Gilmer et al. (2019). Since the primary purpose of MEIB and the other compared methods is not a defense against strong adversarial attacks, we use the Fast Gradient Sign Method (FGSM) Goodfellow et al. (2015):

$$\widetilde{x} = x + \epsilon * \text{sign}(\nabla_x J(\theta, x, y)) \tag{10}$$

where $\widetilde{x}$ is the crafted adversarial example, $\epsilon$ is a scale factor of the perturbations, and $J$ is the target loss function, e.g., cross-entropy loss. The FGSM is often regarded as a weak adversary but is still widely used as a first benchmark adversary method due to its simplicity. We report evaluation results with stronger adversaries in the supplemental Section G for interested readers. We crafted the adversarial examples from the QMNIST test set using the FGSM with $\epsilon \in [0.0, 0.5]$ with the step size of 0.5 on all models ($D = 256$) trained with different random seeds for each method. We used 12 MC samples of $z$ for the stochastic methods, MEIB and VIB.

Figure 3a shows the misclassification rate of each method toward the different strengths of the FGSM perturbations. MEIB is more robust than the other methods with significant gaps. Furthermore, the error rate of MEIB increases very slowly with the increasing strength of perturbations until about $\epsilon = 0.3$. On the other hand, VIB is more vulnerable than both deterministic and dropout baselines, typically with more severe perturbations. It might be because we chose the target model with the best performance with the clean dataset, which is a rational choice, for all methods. Thus, a VIB model trained with a different $\beta$ might yield better robustness, but it still would be difficult to close

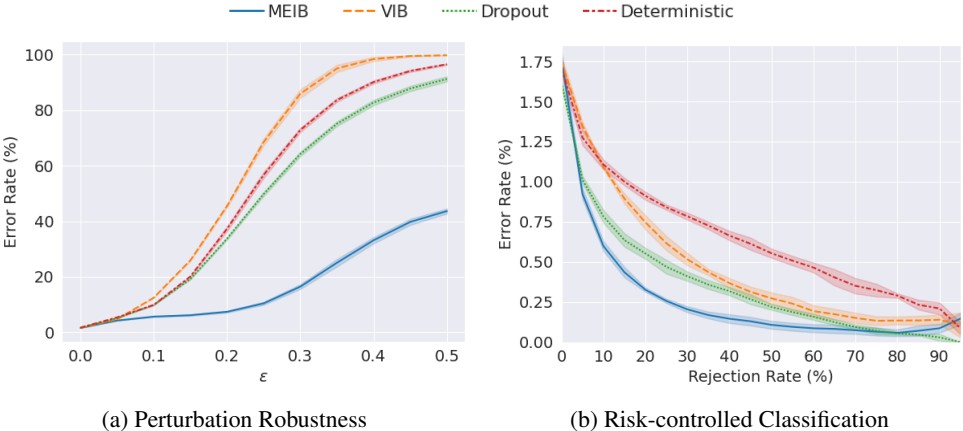

(a) Perturbation Robustness

(b) Risk-controlled Classification

Figure 3: Performance comparisons on the QMNIST dataset.

the gap with MEIB. Please refer to the supplemental Section C for the perturbation robustness with different $\alpha$ values.

**Risk-controlled Classification** In many real-world application scenarios, it would be favorable to refuse any decision instead of making a false prediction when the model is not confident about the input. For doing this, we need a way to correctly estimate the confidence, or the uncertainty, of inputs to ML models. Using the estimated confidence, we may reject the inputs with insufficient confidence; it is called *risk-controlled recognition* Shi & Jain (2019). We evaluated the risk-controlled classification performance on the QMNIST test set by rejecting the inputs by the confidence estimated by each method. Similarly to the confidence measured by MEIB (Section 3), we empirically found that the mean of variance vector from $f_\theta^\Sigma(x)$ of VIB is proportional to the confidence. For the deterministic and dropout baselines, we use the $L_2$-norm of the embedding vector for each input as a proxy measure of confidence Ranjan et al. (2017). We set the rejection rate from $0\%$ to $95\%$ with a step size of $5\%$. A single sample of $z$ was used for MEIB and VIB. Figure 3b shows the risk-controlled classification performance by all methods with $D = 256$. MEIB outperformed all other methods across the most range of the input rejection rate. Specifically, the error rate of MEIB dropped more than half of the initial value after rejecting $10\%$ of uncertain inputs and reached about $0.1\%$ when half of the inputs were rejected.

**Embedding Distribution** Figure 4 depicts the embedding space learned by VIB and MEIB with $D = 2$. It shows that each embedding distribution by MEIB takes as much area as possible depend-

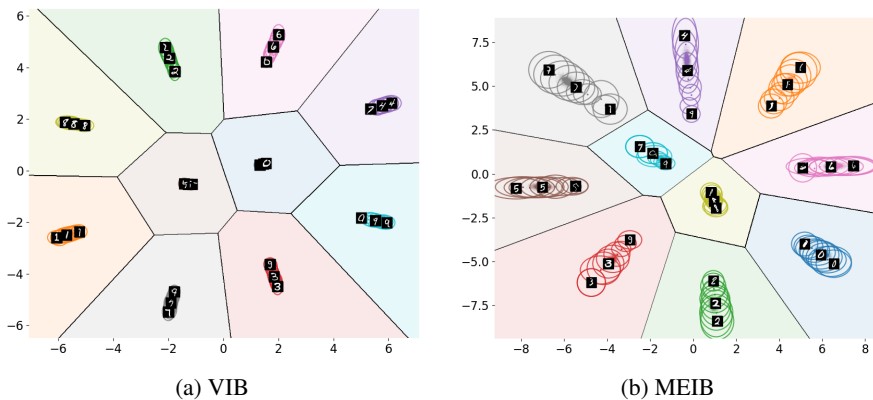

(a) VIB

(b) MEIB

Figure 4: 2D embedding space learned for the QMNIST dataset. The ellipses represent the standard deviation of the stochastic embeddings for a subset of training data.

Table 2: Performance (AP) comparison of HIB with different main framework methods

| $D$ | Test Data | $N = 2$ | | $N = 3$ | |
| --- | --- | --- | --- | --- | --- |
| | | VIB | MEIB | VIB | MEIB |
| 2 | Clean | $0.955 \pm 0.004$ | $\mathbf{0.959 \pm 0.004}$ | $0.950 \pm 0.003$ | $\mathbf{0.954 \pm 0.003}$ |
| | Corrupted | $\mathbf{0.840 \pm 0.004}$ | $0.836 \pm 0.010$ | $\mathbf{0.844 \pm 0.007}$ | $0.842 \pm 0.005$ |
| 3 | Clean | $0.980 \pm 0.001$ | $\mathbf{0.982 \pm 0.002}$ | $0.980 \pm 0.003$ | $\mathbf{0.984 \pm 0.002}$ |
| | Corrupted | $0.861 \pm 0.003$ | $\mathbf{0.864 \pm 0.004}$ | $\mathbf{0.900 \pm 0.006}$ | $0.898 \pm 0.003$ |
| 4 | Clean | $0.991 \pm 0.001$ | $\mathbf{0.993 \pm 0.001}$ | $0.990 \pm 0.002$ | $\mathbf{0.991 \pm 0.001}$ |
| | Corrupted | $0.888 \pm 0.003$ | $\mathbf{0.894 \pm 0.010}$ | $0.912 \pm 0.004$ | $\mathbf{0.913 \pm 0.003}$ |
| 100 | Clean | $\mathbf{1.000 \pm 0.000}$ | $\mathbf{1.000 \pm 0.000}$ | $\mathbf{1.000 \pm 0.000}$ | $\mathbf{1.000 \pm 0.000}$ |
| | Corrupted | $0.931 \pm 0.005$ | $\mathbf{0.936 \pm 0.004}$ | $0.952 \pm 0.004$ | $\mathbf{0.968 \pm 0.002}$ |

ing on its location from the decision boundaries, which is consistent with our hypothesis. On the other hand, every embedding by VIB has a small standard deviation and thus covers a much smaller area in both dimensions (axes) than those of MEIB. It would be reasonable to consider increasing $\sigma$ of the prior distribution $\mathcal{N}(\mathbf{0}, \sigma^2 \mathbf{I})$ used in VIB modeling to achieve a similar effect of entropy maximization by MEIB. However, even VIB with very large variance priors still performs much worse than MEIB for both aspects of perturbation robustness and risk-controlled classification (see the supplemental Section D). It implies that MEIB increases $\sigma$ of inputs in an adaptive way while VIB tries to match the given prior distribution.

## 4.2 Hedged Instance Embedding

Oh et al. (2019) proposed the hedged instance embedding (HIB), a metric learning method that explicitly models the input uncertainty in the stochastic embedding space. HIB utilizes the VIB objective with the probabilistic contrastive learning framework. The authors also proposed a new dataset, called N-digit MNIST, basically images of $N$ adjacent MNIST digits. We examined the HIB framework with the MEIB objective instead of VIB simply by replacing the KL divergence term in the original HIB loss function with the negative conditional entropy of embeddings, keeping all the other aspects of the neural network model same, including the hyperparameters. Please refer to the supplementary material for the architectural details. Table 2 reports the performance of the original HIB with the VIB objective and the MEIB-variant by the average precision (AP) on the test set. MEIB-variant outperformed the original HIB in every case with the clean test set, except the very high-dimensional embedding of $D = 100$, where both performed well. For the corrupted test data, on the other hand, the original HIB performed slightly better in low-dimensional embedding $D = 2$ scenarios, and both performed comparably with $D = 3$. However, MEIB outperformed with a higher-dimensional embedding $D = 4$ and $D = 100$ in both $N = 2$ and 3 cases. In most applications, it is very uncommon to use such a small size of embeddings with two or three dimensions, except for direct visualization of data relationships An et al. (2020). Consequently, it suggests that the MEIB objective would better fit the contrastive learning framework of HIB than the VIB objective upon using a reasonable size of embedding dimensions.

## 4.3 Person Re-identification

Person re-identification (ReID) is an important computer vision task that is utilized in various applications, including intelligent security and surveillance systems Gong et al. (2011). Unlike typical image classification tasks, the objective of the person ReID task is to find the ranked matches of pedestrian images captured across multiple non-overlapping cameras Chen et al. (2021). Person ReID is challenging due to image-level corruptions and appearance changes, including occlusions Ye et al. (2021). Therefore, it is critical to have an embedding method robust to noises and capable of confidence measuring for potential risk-controllability. For evaluating those aspects, we used three datasets popularly employed in person ReID literature: Market-1501Zheng et al. (2015), MSMT17Wei et al. (2018), and LPW Song et al. (2018). For LPW, we used a quarter subset of it by selecting every fourth frame of each identity due to the limited computational resource.

Table 3: Performance on the person ReID datasets

| Dataset | Method | mAP (%) | Rank-1 (%) | Rank-5 (%) | Rank-10 (%) |
|---|---|---|---|---|---|
| Market-1501 | Baseline | $75.65 \pm 0.21$ | $90.48 \pm 0.45$ | $96.47 \pm 0.06$ | $97.66 \pm 0.15$ |
| | DistNet | $74.61 \pm 0.06$ | $90.10 \pm 0.29$ | $96.25 \pm 0.23$ | $97.61 \pm 0.26$ |
| | PFE | $76.49 \pm 0.23$ | $90.48 \pm 0.51$ | $96.53 \pm 0.11$ | $97.75 \pm 0.20$ |
| | DUL | $77.12 \pm 0.19$ | $90.09 \pm 0.17$ | $95.84 \pm 0.15$ | $97.29 \pm 0.18$ |
| | MEIB | $\mathbf{79.67 \pm 0.15}$ | $\mathbf{92.14 \pm 0.16}$ | $\mathbf{96.79 \pm 0.16}$ | $\mathbf{97.83 \pm 0.09}$ |
| MSMT17 | Baseline | $39.69 \pm 0.16$ | $69.31 \pm 0.31$ | $82.66 \pm 0.17$ | $86.73 \pm 0.06$ |
| | DistNet | $38.16 \pm 0.16$ | $68.79 \pm 0.21$ | $82.44 \pm 0.32$ | $86.67 \pm 0.28$ |
| | PFE | $42.72 \pm 0.14$ | $70.62 \pm 0.24$ | $83.68 \pm 0.17$ | $87.50 \pm 0.20$ |
| | DUL | $38.70 \pm 0.18$ | $67.28 \pm 0.22$ | $80.98 \pm 0.50$ | $85.38 \pm 0.25$ |
| | MEIB | $\mathbf{44.77 \pm 0.39}$ | $\mathbf{73.85 \pm 0.41}$ | $\mathbf{85.34 \pm 0.25}$ | $\mathbf{88.80 \pm 0.24}$ |
| LPW | Baseline | $34.82 \pm 0.33$ | $52.76 \pm 0.80$ | $67.06 \pm 0.94$ | $73.32 \pm 0.87$ |
| | DistNet | $32.37 \pm 0.29$ | $50.25 \pm 0.20$ | $65.36 \pm 0.57$ | $72.02 \pm 0.89$ |
| | PFE | $33.35 \pm 0.34$ | $50.36 \pm 0.41$ | $64.99 \pm 0.67$ | $71.69 \pm 0.76$ |
| | DUL | $33.41 \pm 0.65$ | $49.41 \pm 0.90$ | $63.90 \pm 0.83$ | $70.13 \pm 0.83$ |
| | MEIB | $\mathbf{37.64 \pm 0.41}$ | $\mathbf{54.85 \pm 0.52}$ | $\mathbf{69.05 \pm 0.58}$ | $\mathbf{75.20 \pm 0.49}$ |

For all methods compared in ReID experiments, we used ResNet50 He et al. (2016) as the backbone encoder $f_\theta^B$. The mean estimator $f_\theta^\mu$ of an FC layer with 512 hidden units followed by a BN layer was used for all methods. For the variance estimator $f_\theta^\Sigma$, we used an MLP of FC512-BN-ReLU-FC512-exp for MEIB and followed the original architecture proposed by the authors for each method, except that we used 512 hidden units for their FC layers. We used $\alpha = 1$ for MEIB and the recommended hyperparameter values by the authors for each method.

We compare the performance of MEIB with the following baseline methods not only from person ReID literature but also from previous work on face recognition, which shares similar task characteristics: DistributionNet (DistNet) Yu et al. (2019), Probabilistic Face Embedding (PFE) Shi & Jain (2019), and Data Uncertainty Learning (DUL, equivalent to VIB) Chang et al. (2020). In addition, we compare the deterministic baseline of the same architecture without the variance estimator $f_\theta^\Sigma$. We also tried HIB for the ReID task, but we could not achieve meaningful results. All models were trained for the classification task with a softmax classifier of $f_\theta^C$ to predict the true identity labels. In testing time, the Euclidean distance between every pair of gallery and query images is calculated using the deterministic embeddings or the mean of stochastic embeddings $f_\theta^\mu(x)$ of them to rank the pairs. For PFE, we use the negative mutual likelihood score (MLS) used in the original work of PFE as the distance metric.

We first evaluated the methods without considering the input confidence and risk control. Table 3 summarizes the results where we report the mean average precision (mAP) Schütze et al. (2008) and cumulative matching characteristics (CMC) curve Gray et al. (2007) at rank-1, rank-5, and rank-10. MEIB outperforms all the others by from 2 up to 6.6 percentage points of mAP throughout the datasets considered. It implies that maximizing the conditional entropy of embeddings in MEIB has better regularization effects than the other methods.

The importance of risk-controlled recognition is more emphasized in the person ReID task due to the cost of misidentifying a person; it can have a serious societal impact, such as falsely tagging someone as a criminal. Considering a real-world application of risk-controlled ReID model deployment, a realistic situation is that we can prepare a well-curated clean set of gallery images in advance and expect that arbitrary query/probe images with potential corruptions will be given after model deployment. Thus, we kept the original test gallery images for each dataset and evaluated the methods with varying amounts of rejected test query images. The query images are sorted using the confidence estimated by each method: MEIB and the deterministic model use the same approaches described in Section 4.1. DistNet and DUL use the mean of variance vector from $f_\theta^\Sigma(x)$, while PFE uses the mean reciprocals of the variance vector elements as their confidence measures. Then, the first $R\%$ of low-confidence images are removed from evaluation. For each remaining query image, the entire gallery images are ranked by the distance, and the evaluation metrics are calculated.

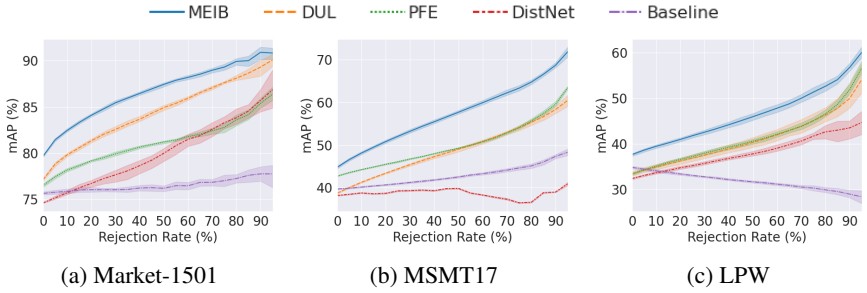

(a) Market-1501    (b) MSMT17    (c) LPW

Figure 5: Risk-controlled person ReID performance.

Figure 5 shows the risk-controlled identification performance of the methods for each dataset. While MEIB starts with the best performance among all methods, as shown previously, the amount of performance improvement at lower rejection rates is more significant than the other methods in most cases. Furthermore, the gap between MEIB and the other methods is kept over the most range of the rejection rate. It confirms that MEIB provides an effective confidence measure and risk-control capability. Figure 6 shows the example query images for each dataset (except MSMT17 prohibited by the license) with the associated confidence (conditional entropy) value. There are conspicuous trends: (1) the low-confident images are mostly blurry while the high-confident images are relatively clearer, (2) the persons in the low-confident images are often occluded by another object or person, and (3) the most persons in the low-confident images wear clothes with an achromatic color without patterns while those in the high-confident images wear clothes with vivid colors or patterns such as stripes. This would be another evidence that MEIB provides reasonable confidence measurements.

## 5 CONCLUSION

In this work, we presented MEIB, a novel framework that produces stochastic embeddings distributed with the maximum entropy. MEIB provides a regularization effect and robustness toward adversarial noise by securing the maximum area in the embedding space, leading to better classification performance. Moreover, the experimental results in digit classification and person ReID tasks showed that MEIB enables an effective risk-controlled recognition by providing reliable confidence measurements. While the experiments in this work utilized somewhat elementary neural network backbone encoders, it would be straightforward to combine MEIB with more sophisticated architectures to yield further classification performance improvement.

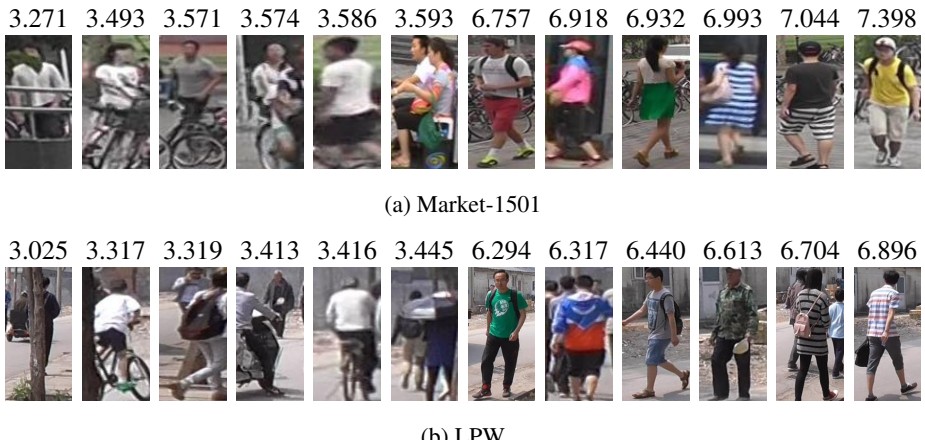

Figure 6: Example query images that MEIB is least confident (left six) and most confident (right six) from each ReID dataset. The number above each image is the confidence measured by MEIB as the dimension-wise average entropy.

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

## A    UNCERTAINTY ESTIMATION OF VIB

Figure 7 shows the QMNIST test set samples with the lowest and highest variance averaged over feature dimension, i.e., $\frac{1}{D} \sum_{d=1}^{D} \sigma_{\theta,d}^2$, estimated by VIB. It shows that VIB assigns relatively smaller variances for uncertain inputs and higher variances for clear inputs. Moreover, Figure 8 shows the risk-controlled classification (see Section 4.1 in the main paper) performance of VIB with the ascending and descending order of the average variance. In the ascending order, the samples with small variances are rejected first. Similarly, the high variance samples are rejected first in the descending order case. If we assume that the variance proportionally represents the 'uncertainty' associated with the inputs, we will reject the samples with the higher variance first. In this case of the descending order rejection, the risk-controlled classification did not work as expected; the more samples were rejected, the worse performance was achieved. On the contrary, the risk-controlled classification worked correctly using the ascending order, regardless of its performance. Therefore, we can infer from these results that it is more rational to utilize the variance $\sigma_\theta^2$ in VIB to represent a 'confident' area/interval rather than the 'uncertainty'.

0.584   0.613   0.625   0.629   0.632   0.634   0.636   0.639   0.640   0.642   0.643   0.648

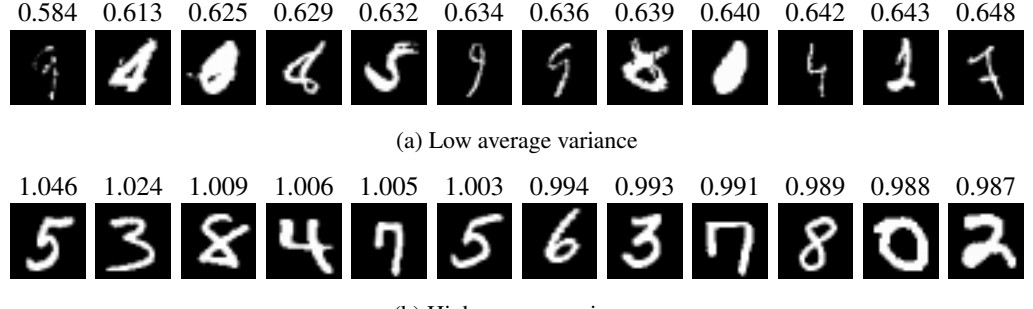

(a) Low average variance

1.046   1.024   1.009   1.006   1.005   1.003   0.994   0.993   0.991   0.989   0.988   0.987

(b) High average variance

Figure 7: Sample QMNIST images with the uncertainty estimated by VIB.

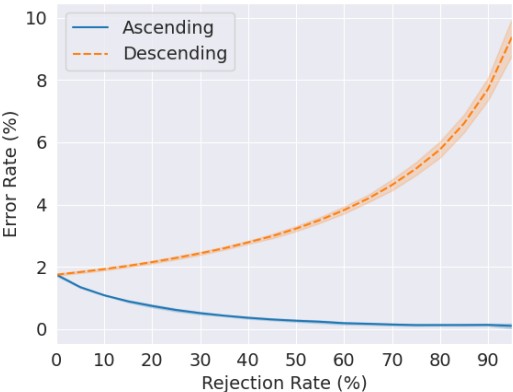

Figure 8: Risk-controlled classification performance of VIB with the ascending and descending order of the average variance.

# B   CROSS-ENTROPY $H(p_\theta(z|x_n), r(z))$

Let two $D$-dimensional multivariate Gaussian distributions $p_\theta(z|x_n) = \mathcal{N}(z|\boldsymbol{\mu}_\theta, \Sigma_\theta)$ and $r(z) = N(z; \boldsymbol{\mu_0}, \Sigma_\mathbf{I})$ where $\boldsymbol{\mu}_\theta = f_\theta^{\boldsymbol{\mu}}(x_n)$, $\Sigma_\theta = \boldsymbol{\sigma}_\theta^2 \mathbf{I} = f_\theta^\Sigma(x_n)$, a diagonal covariance, $\boldsymbol{\mu_0} = \mathbf{0}$, a zero mean vector, and $\Sigma_\mathbf{I} = \mathbf{I}$, an identity covariance matrix, respectively. Then, $H(p(z|x_n), r(z))$, the cross-entropy of $r(z)$ relative to $p(z|x_n)$, is given by

$$H(p(z|x_n), r(z)) = -\int p(z|x_n) \ln r(z) \, dz \tag{11}$$

$$= \frac{1}{2} \int \mathcal{N}(z|\boldsymbol{\mu}_\theta, \Sigma_\theta) \Big( D \ln(2\pi) + \ln|\Sigma_\mathbf{I}| + (z - \boldsymbol{\mu_0})^\top \Sigma_\mathbf{I}^{-1}(z - \boldsymbol{\mu_0}) \Big) \, dz \tag{12}$$

$$= \frac{1}{2} \int \mathcal{N}(z|\boldsymbol{\mu}_\theta, \Sigma_\theta) \Big( D \ln(2\pi) + \ln|\mathbf{I}| + (z - \mathbf{0})^\top \mathbf{I}^{-1}(z - \mathbf{0}) \Big) \, dz \tag{13}$$

$$= \frac{1}{2} \int \mathcal{N}(z|\boldsymbol{\mu}_\theta, \Sigma_\theta) \Big( D \ln(2\pi) + z^\top z \Big) \, dz \tag{14}$$

$$= \frac{1}{2} \left( D \ln(2\pi) \int \mathcal{N}(z|\boldsymbol{\mu}_\theta, \Sigma_\theta) \, dz + \int \mathcal{N}(z|\boldsymbol{\mu}_\theta, \Sigma_\theta)(z^\top z) \, dz \right) \tag{15}$$

$$= \frac{1}{2} \left( D \ln(2\pi) + \mathbb{E}_{z \sim p_\theta(z|x_n)} \left[ z^\top z \right] \right) \tag{16}$$

$$= \frac{1}{2} \left( D \ln(2\pi) + \mathbb{E}_{z \sim p_\theta(z|x_n)} \left[ \sum_{d=1}^{D} z_d^2 \right] \right) \tag{17}$$

$$= \frac{1}{2} \left( D \ln(2\pi) + \sum_{d=1}^{D} \mathbb{E}_{z \sim p_\theta(z|x_n)} \left[ z_d^2 \right] \right) \tag{18}$$

$$= \frac{1}{2} \left( D \ln(2\pi) + \sum_{d=1}^{D} \left( \mathbb{E}_{z \sim p_\theta(z|x_n)} \left[ z_d \right]^2 + \mathrm{Var}_{z \sim p_\theta(z|x_n)} \left[ z_d \right] \right) \right) \tag{19}$$

$$= \frac{1}{2} \left( D \ln(2\pi) + \sum_{d=1}^{D} \left( \boldsymbol{\mu}_{\theta,d}^2 + \boldsymbol{\sigma}_{\theta,d}^2 \right) \right) \geq 0 \tag{20}$$

where each of $z_d$, $\boldsymbol{\mu}_{\theta,d}$, and $\boldsymbol{\sigma}_{\theta,d}^2$ is the $d$-th dimension of the corresponding vector. Note again that $\Sigma_\theta = \boldsymbol{\sigma}_\theta^2 \mathbf{I} = f_\theta^\Sigma(x_n)$, a diagonal covariance, i.e., each dimension of $z$ is independent to each other.

# C   PERTURBATION ROBUSTNESS BY DIFFERENT $\alpha$

Figure 9a shows the misclassification rates against the FGSM perturbations for MEIB trained with different $\alpha$ values. It shows a more consistent trend that a larger $\alpha$ provides more robustness for the given perturbation strength, unlike the VIB case that shows saddle-shaped curves in Figure 9b. It is reasonable because a larger $\alpha$ (thus, a larger $\beta$) value encourages the model to assign larger entropy and thus secure a larger area for the distribution of $z$ given input $x$, which leads to larger margins from the decision boundary.

# D   VIB WITH LARGE-VARIANCE PRIORS

We trained and evaluated VIB models with different values of $\boldsymbol{\sigma}$ in their prior distribution. Figure 10 shows the embeddings of VIBs trained with 2-dimensional bottleneck and different values of $\boldsymbol{\sigma}$. By visual inspection, it may seem getting similar to those of MEIB shown in Figure 4b as $\boldsymbol{\sigma}$ increases. On the other hand, Figure 11 presents the perturbation robustness (toward FGSM attacks) and risk-controlled classification performance by VIB trained with the different values of $\boldsymbol{\sigma}$ while keeping the other configurations same as in Section 4, including the 256-dimensional bottleneck. It shows that the adversarial robustness is improved with increasing $\boldsymbol{\sigma}$, until $\boldsymbol{\sigma} = 10$, but becomes worse again with much larger $\boldsymbol{\sigma} = 100$, while all of them were still worse than MEIB with a significant

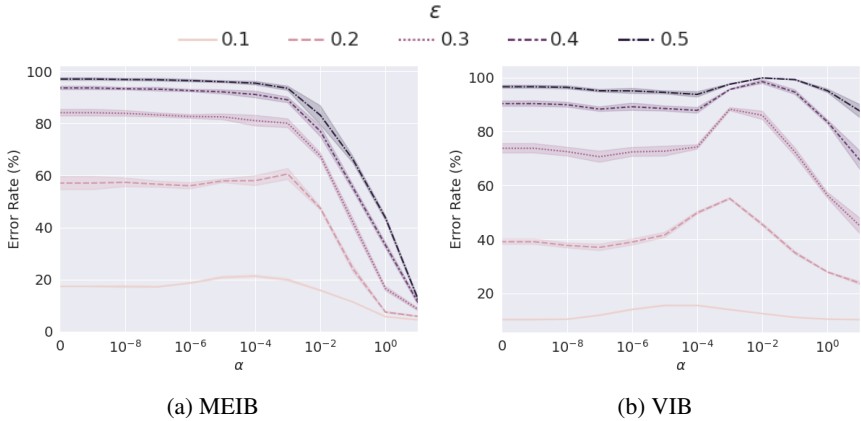

(a) MEIB        (b) VIB

Figure 9: Adversarial robustness with different $\alpha$ values.

gap. Moreover, the risk-controlled classification performance has little difference with different $\boldsymbol{\sigma}$ values.

On the other hand, the magnitude of $\boldsymbol{\sigma}$s estimated by MEIB for QMNIST dataset are in a range below than those by VIB with $\boldsymbol{\sigma} = 5$, as shown in Table 4. It provides an insight that MEIB increase $\sigma$ of inputs in a more effective way.

Table 4: $\| \boldsymbol{\sigma} \|_2$ estimated by each method

| Method | Mean | SD | Min | Max |
|---|---|---|---|---|
| VIB ($\boldsymbol{\sigma} = 1$) | 0.38 | 0.08 | 0.18 | 0.53 |
| VIB ($\boldsymbol{\sigma} = 5$) | 1.80 | 0.39 | 0.56 | 4.03 |
| VIB ($\boldsymbol{\sigma} = 10$) | 3.22 | 0.84 | 0.70 | 9.33 |
| VIB ($\boldsymbol{\sigma} = 100$) | 18.77 | 8.01 | 2.04 | 81.42 |
| MEIB | 1.10 | 0.28 | 0.45 | 1.95 |

## E    RANK-1 ACCURACY FOR RISK-CONTROLLED PERSON REID

Figure 12 shows the risk-controlled CMC rank-1 accuracy for each method on each ReID dataset. The results were obtained from the same experiments described in Section 4.3, in addition to the risk-controlled mAP in Figure 5 of the main paper. It confirms again that MEIB provides the most effective confidence measure and risk-control capability compared to the other methods over the most range of the rejection rate.

## F    HYPERPARAMETER STUDY

Figure 13 shows the performance of VIB and MEIB for different values of $\alpha$ and the usage of the BN layer at the end of the $f_\theta^\mu$ branch in the QMNIST digit classification task with $D = 256$. For VIB, the BN layer is beneficial only for the small values of $\alpha$ ($0 \sim 10^{-4}$), and the best result was achieved without using the BN layer at $\alpha = 0.01$. For MEIB, on the other hand, it is a common phenomenon that using the BN layer at the end of $f_\theta^\mu$ noticeably improves the performance across all $\alpha$ values, while the best performance was obtained with $\alpha = 1$ using the BN layer. Furthermore, Figure 14 shows that using the BN layer at the end of the $f_\theta^\mu$ branch also significantly improves the performance of MEIB on all datasets considered in the person ReID task, commonly with the best performance at $\alpha = 1$. While we need a more concrete study about the effect of BN in MEIB, which we leave as our future work, it can be assumed that $\alpha = 1$ with a BN layer at the end of $f_\theta^\mu$ is a reasonable default configuration for these tasks.

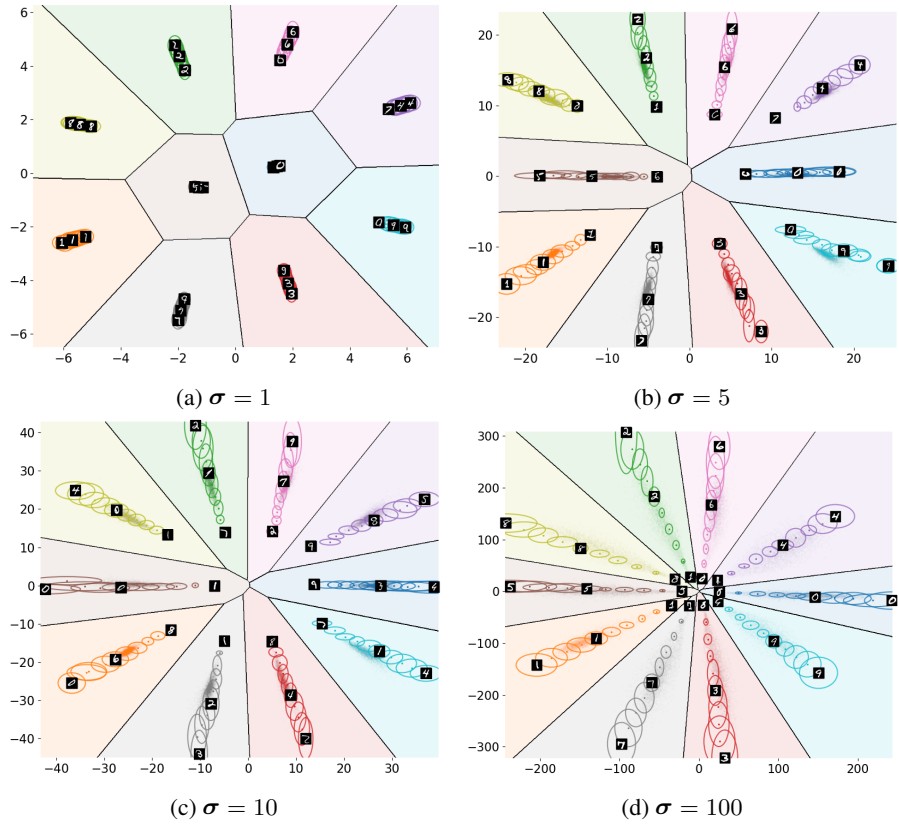

(a) $\sigma = 1$

(b) $\sigma = 5$

(c) $\sigma = 10$

(d) $\sigma = 100$

Figure 10: 2D embedding space learned for the QMNIST dataset by VIB with different $\sigma$ of the prior distributions. The ellipses represent the standard deviation of the stochastic embeddings for a subset of training data.

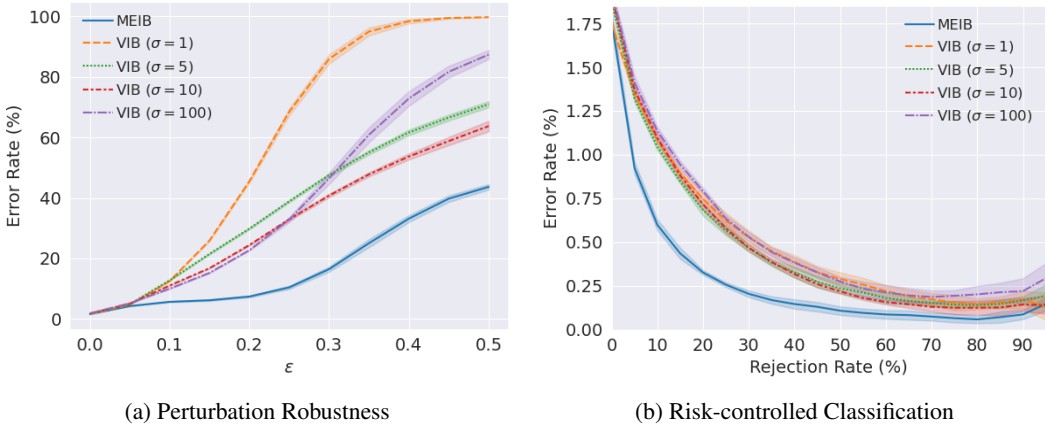

(a) Perturbation Robustness

(b) Risk-controlled Classification

Figure 11: Performance comparisons on the QMNIST dataset.

## G   STRONGER ADVERSARIAL ATTACKS EXPERIMENTS

While the FGSM evaluated in Section 4 is a popular first-choice baseline method, it is often ineffective compared to multi-step attack methods. Although the main contribution of this paper is not an adversarial defensive method, we evaluated MEIB and other baseline methods on the following stronger adversaries for a more concrete adversarial robustness analysis: the multi-step Projected Gradient Descent (PGD) Madry et al. (2018), two variants of Auto-PGD (APGD), one based on the

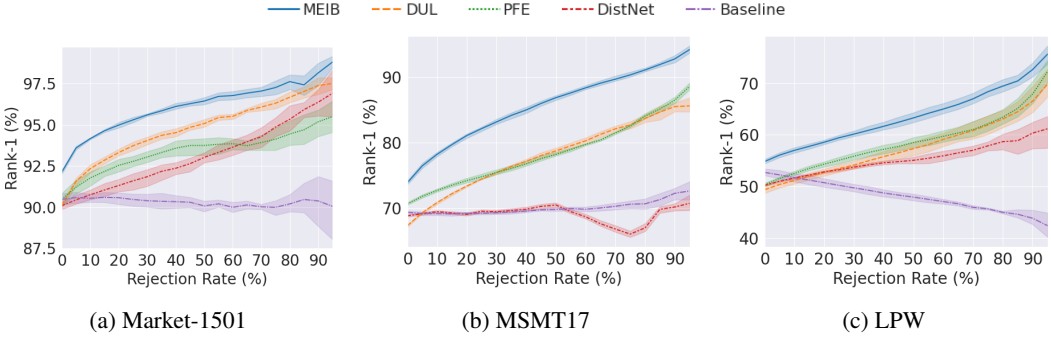

(a) Market-1501      (b) MSMT17      (c) LPW

Figure 12: CMC rank-1 accuracy for risk-controlled person ReID.

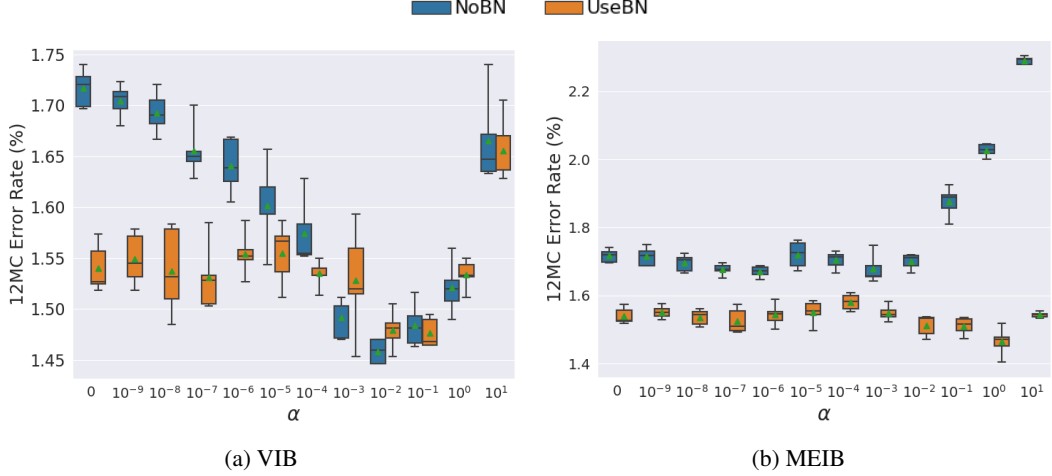

(a) VIB      (b) MEIB

Figure 13: Hyperparameter comparison for QMNIST (lower is better).

cross-entropy loss ($APGD_{CE}$) and one with the difference of logits ratio (DLR) loss ($APGD_{DLR}$), and AutoAttack (AA), which is an ensemble of those two APGDs Croce & Hein (2020). We used the projection on the $L_\infty$-ball of radius $\epsilon$ for all attacks where we set $\epsilon = 0.3$. For PGD, we used 1000 steps (PGD-1000) with the step size $\epsilon_{step} = 0.01$ and 100 random restarts. For APGDs and AA, Expectation over Transformation (EOT) Athalye et al. (2018) was used with an average over 20 times of the gradient computations for more effective attacks on the stochastic models, MEIB and VIB. We used Adversarial Robustness Toolbox (ART) Nicolae et al. (2018) to leverage its PGD implementation and the AA implementation provided by the authors[2], including APGDs.

Table 5 shows the evaluation results on class-balanced $10,000$ samples of the QMNIST test set. MEIB outperformed all the other baseline methods across all different types of adversarial attacks. Although MEIB was also effectively fooled by these multi-step and adaptive PGD attacks, it performs better than other non-adversarially trained models such as Mixup evaluated with the PGD attacks on the original MNIST datasetAddepalli et al. (2020). It would be possible to achieve better robustness once MEIB is trained with adversarial training methods.

## H IMPLEMENTATION AND TRAINING DETAILS

**QMNIST Digit Classification** All models were trained for 200 epochs with a batch size of 100. Adam optimizer Kingma & Ba (2015) was used with $\beta_1 = 0.5, \beta_2 = 0.999$, and the learning rate of 0.0001, which is decayed by the factor of 0.97 every other epoch. The exponential moving average of the model parameters with a decay factor of 0.999 was tracked during the training, and the final

---

[2]https://github.com/fra31/auto-attack

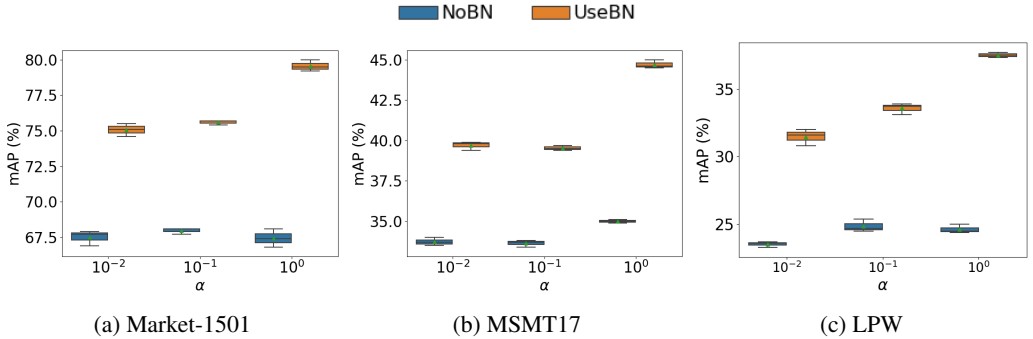

(a) Market-1501  (b) MSMT17  (c) LPW

Figure 14: MEIB hyperparameter comparison for person ReID (higher is better).

Table 5: Classification Accuracy (%) with different adversarial attacks on the QMNIST test samples

| Method | Clean | PGD-1000 | APGD$_{\text{CE}}$ | APGD$_{\text{DLR}}$ | AA |
|---|---|---|---|---|---|
| Deterministic | $98.39 \pm 0.11$ | $1.13 \pm 0.04$ | $0.00 \pm 0.00$ | $0.00 \pm 0.00$ | $0.00 \pm 0.00$ |
| Dropout | $\mathbf{98.45 \pm 0.08}$ | $1.09 \pm 0.04$ | $0.01 \pm 0.01$ | $0.00 \pm 0.00$ | $0.00 \pm 0.00$ |
| VIB | $98.34 \pm 0.14$ | $1.18 \pm 0.07$ | $0.89 \pm 0.05$ | $1.20 \pm 0.10$ | $0.06 \pm 0.02$ |
| MEIB | $98.38 \pm 0.15$ | $\mathbf{2.42 \pm 0.44}$ | $\mathbf{6.79 \pm 1.40}$ | $\mathbf{1.60 \pm 0.28}$ | $\mathbf{0.12 \pm 0.06}$ |

averaged parameters were used at test time. We trained and evaluated all methods with five different random seed values. All pixel values of input digit images were rescaled to $[-1, 1]$. The embedding size of $D = 256$ was used by default for all experiments unless it was specified. For the stochastic methods, MEIB and VIB, we used 12 MC samples of $z$ for the adversarial robustness experiment and a single sample for the risk-controlled classification experiment.

**Hedged Instance Embedding**  We used the same architecture in the original HIB work Oh et al. (2019), except for some parts not mentioned by the authors. The backbone encoder $f_\theta^B$ consists of two convolutional layers, 32 and 64 filters of $5 \times 5$ kernels, each followed by ReLU activation and a max-pooling with $2 \times 2$ kernels. Each $f_\theta^{\boldsymbol{\mu}}$ and $f_\theta^{\Sigma}$ is an FC layer of $D$ hidden units, and a BN layer is attached at the end of $f_\theta^{\boldsymbol{\mu}}$. Both MEIB and VIB variants of the HIB models were trained over 500,000 iterations with a batch size of 128. We followed the same batching strategy used in Oh et al. (2019) to ensure we had enough number of both positive and negative pairs of images. Adam optimizer Kingma & Ba (2015) was used with $\beta_1 = 0.5, \beta_2 = 0.999$ and the learning rate of 0.0001.

**Person Re-identification**  We utilized the Torchreid framework Zhou & Xiang (2019) to implement the methods and conduct the experiments. Except for PFE and DistNet, all other models were initialized with the ResNet50 parameters pre-trained on the ImageNet dataset Deng et al. (2009) and trained for 60 epochs with a batch size of 32. Specifically, we adopted the two-stepped transfer learning strategy Geng et al. (2016), where the backbone encoders $f_\theta^B$ were frozen for the first 5 epochs. AMSGrad optimizer Reddi et al. (2018) was used with the learning rate of 0.0003, which was reduced by the factor of 10 every 20 epochs. PFE and DistNet were initialized from the fully-trained deterministic baseline model and fine-tuned over another 60 epochs only for the parts specified in their original work. While the same optimizer was used, the initial learning rate was set as 0.0001, which was also reduced by the factor of 10 every 20 epochs. For all methods, each batch consists of 4 random identities and 8 random images for each identity. All input images were rescaled to $128 \times 256$, and random horizontal flipping with the probability of 0.5 was applied. We reported the mean and standard deviation of the performance for all models trained with five different random seeds.

## I  MISCLASSIFIED QMNIST SAMPLES

Figure 15 shows a subset of QMNIST test images misclassified by MEIB after filter out $80\%$ of the test set in the risk-controlled classification experiment in Section 4.1. Although MEIB was overconfident, i.e., assigned inappropriately high entropy values, for these uncertain images, each image has a plausible shape for the predicted class also.

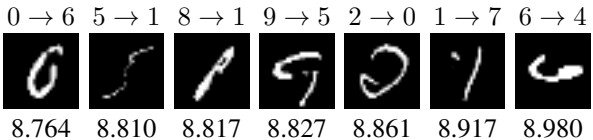

$0 \to 6 \quad 5 \to 1 \quad 8 \to 1 \quad 9 \to 5 \quad 2 \to 0 \quad 1 \to 7 \quad 6 \to 4$

8.764  8.810  8.817  8.827  8.861  8.917  8.980

Figure 15: Sample QMNIST images misclassified by MEIB. Two digits above each image represent `Label →Prediction` and the number below each image is the conditional entropy estimated.

## J  L2 NORMALIZATION FOR MEIB AND VIB

$L_2$ normalization of feature embedding Ranjan et al. (2017) is widely used with the softmax loss Wang et al. (2017), angular margin-based loss Deng et al. (2019), and contrastive loss Chen et al. (2020). We also trained and tested MEIB and VIB with $L_2$-normalized $z$ samples to see its benefits on stochastic embeddings. Figure 16 shows the perturbation robustness and risk-controlled classification performance, similar to Figure 3, of the MEIB and VIB models with and without $L_2$ normalization of embeddings. MEIB performs much worse with the $L_2$ normalization in terms of both perturbation robustness and risk-controlled classification. While VIB shows a slightly improved robustness with $L_2$ normalization, its risk-controlled classification performance become worse than without the normalization shortly after about 30% of rejection rate. It could be difficult for stochastic embeddings, especially MEIB, to utilize their advantage in angular space since their magnitudes are not considered anymore. For example, the main intuition of MEIB is spreading out each input embedding as wide as possible to take more space within a decision region. However, after applying $L_2$ normalization, only the angle of each feature vector contributes to the decision regardless of their magnitude in the embedding space.

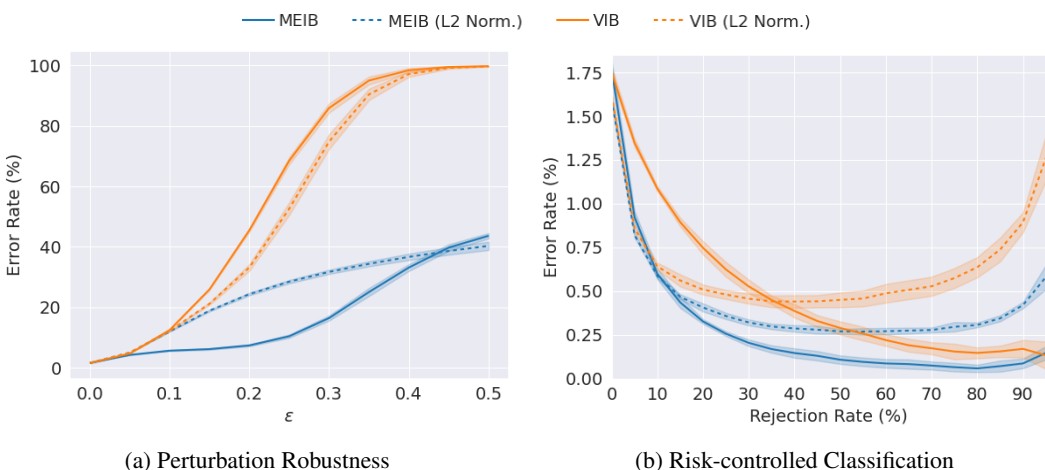

(a) Perturbation Robustness  (b) Risk-controlled Classification

Figure 16: Comparison of $L_2$-normalized MEIB and VIB on the QMNIST dataset.

