# OpenReview forum: "Maximum Entropy Information Bottleneck for Confidence-aware Stochastic Embedding"
_ICLR.cc/2023/Conference — Submitted to ICLR 2023_

### Official Review · Reviewer_PNBE · 2022-10-17

**Confidence:** 4
**Correctness:** 3
**Technical Novelty And Significance:** 2
**Empirical Novelty And Significance:** 3
**Recommendation:** 3

**Clarity, Quality, Novelty And Reproducibility:**

## Clarity and Reproducibility
The paper is generally clearly written, with a few exceptions that focus, unfortunately, on the experimental section. Training details are given in the supplementary material.
- The fundamental difference to VIB is not entirely clear. Essentially, it is based on removing the term $H(Z)$ from $I(X;Z)$, maximizing only the conditional entropy rather than minimizing mutual information. (Matching the distribution of $Z$ to a standard Gaussian is only a consequence of the variational approach, not its goal.) Can you explain what the effect of this modification is? And how is this connected to the deterministic IB, where the term $H(Z|X)$ is removed? A better understanding of the effect these terms have on optimization would allow us to interpret MEIB better in the light of the literature.
- On page 2, the first paragraph criticizes VAE and VIB to assign smaller variance to more uncertain inputs (see, e.g., the beginning of the next paragraph: "lift such constraints"). Indeed, it is later shown that MEIB also assigns smaller variance (lower entropy) to more uncertain inputs. Please explain how this is fundamentally different from VIB, and how MEIB now "lifts a constraint".
- HIB by Oh et al. is not explained well. To better understand the effect of exchanging the VIB by the MEIB objective, their role in HIB should be clear.

## Novelty
The MEIB objective is novel to the best of my knowledge, even if it is just a minor, and thus incremental change when compared to VIB or similar works.

## Quality
The experimental evidence is large, covering various datasets, comparison methods, and hyperparameter settings. Results look very much in favor of MEIB. However, some of the choices made are not fully clear, and I would appreciate more information to accept the experimental evidence as fair.
- Regarding the choice of QMNIST, it is not clear why a larger test set is required. Please comment and/or also add results about MNIST in the supplementary material.
- The authors perform an ablation study in Appendix F, where it is shown that BN is essential to achieve good performance. Insights into why this is the case would be highly appreciated (and the authors themselves note that this is future work).
- In Sec. 4.1, certain choices of $\alpha$ were made and kept fixed for all experiments. However, there is an inherent trade-off between adversarial robustness and classification accuracy, and the authors themselves admit that VIB may be more robust for larger values of $\alpha$/$\beta$. The authors should either select $\alpha$ differently for each task, or argue that MEIB is less sensitive to hyperparameter settings, which itself is a great advantage over VIB.
- The results depicted in Fig. 4 will depend strongly on $\beta$ (as shown in the original paper of VIB), and less on the variance $\sigma^2$ of the Gaussian $r(z)$.
- In Sec. 4.2, it is premature to talk about a trend, looking only at $D=2,3,4$. I would appreciate additional experiments with $D=100$ (or similar) to confirm the statements made in the paper.
- In Fig. 9, should not at $\alpha=0$ the performance of VIB and MEIB be identical? Can you explain the difference?

## Minor
- Why is the method called "maximum entropy information bottleneck" and not just "maximum entropy bottleneck"? After all, instead of (mutual) information, now the entropy is used for regularization.
- On the first page, the authors write "This [...] impels the embeddings to be close to a standard normal distribution, which is an explicit assumption that may not always hold true." Indeed, there are formulations of VIB that use more powerful priors $r(z)$, such as GMMs with learned parameters. Can you comment on how MEIB will perform in comparison to these?
- Page 4: $\sigma_\theta^2 \mathbf{I}$ is misleading, as it suggests that all elements on the main diagonal have the same variance.
- Reference Thomas & Joy (2006) is broken.

**Strength And Weaknesses:**

The paper is generally well-written and the math is easy to follow. The cost function is well motivated and its connection with existing methods (such as VIB) is explained. The paper is heavy on the experimental side, which I greatly appreciate. The experimental evidence is good (but see also below), and the supplementary material complements the paper nicely.

Regarding the main weaknesses, I have to admit that the experimental setup and choices are not always entirely clear and that thus it is difficult to judge the significance of the experimental results. Further, some connections to the VIB and related methods are not fully clear and need to be expanded upon.

**Summary Of The Paper:**

The paper proposes a novel cost function for representation learning/stochastic embedding of data. Specifically, they adapt the information bottleneck functional, which aims at maximizing $I(Y;Z)-\beta I(X;Z)$ by removing the term corresponding to the entropy of $Z$. Essentially, the authors result at maximizing $I(Y;Z)+\beta H(Z|X)$, where the former term ensures that the representation $Z$ is useful for the downstream task, and where the latter maximizes the stochasticity of the representation given the input. The authors show that their approach results in a higher adversarial robustness and risk-controlled recognition performance.

**Summary Of The Review:**

The paper is well-written and proposes an interesting adaption of VIB with small, but non-neglibible novelty. The experimental evidence is good, but not fully convincing because some of the experimental setups are not clear or well-justified. Regarding the theoretical argumentations, I would appreciate a more in-depth discussion of the effects of $H(Z)$, $H(Z|X)$, and $I(X;Z)$ as regularization terms, and how MEIB builds on them.

*EDIT*: After discussion with other reviewers, I think that the paper is not sufficiently novel to merit publication. I lowered my score accordingly.

---

> ### Author Response · Authors · 2022-11-19
> **Response to Review**
>
> **Thank you for your time in reviewing our manuscript and your valuable comments. Please find answers to some of the questions posed.**
>
> - On page 2, the first paragraph criticizes VAE and VIB to assign smaller variance to more uncertain inputs (see, e.g., the beginning of the next paragraph: "lift such constraints"). Indeed, it is later shown that MEIB also assigns smaller variance (lower entropy) to more uncertain inputs. Please explain how this is fundamentally different from VIB, and how MEIB now "lifts a constraint".
>
> **We apologize if the statement was confusing. What we meant by “constraints” was to use a fixed prior, not the relationship between the variance and the input uncertainty. We have modified the statement to deliver our idea more clearly.**
>
> - Regarding the choice of QMNIST, it is not clear why a larger test set is required. Please comment and/or also add results about MNIST in the supplementary material.
>
> **Similar to what the authors of the QMNIST dataset pointed out (Yadav & Bottou, 2019), we also had a concern that the 10k test images of MNIST dataset could be too small to evaluate the generalizability of models and that there is a chance to overfit to the small test set. Since the (almost identical) MNIST test set is a subset of the QMNIST test set, we think the results from QMNIST serves as a better evaluation for the generalizability of the models.**
>
> - In Sec. 4.2, it is premature to talk about a trend, looking only at D = 2, 3, 4. I would appreciate additional experiments with D = 100 (or similar) to confirm the statements made in the paper.
>
> **To validate the case with higher dimension, we have added the result with D=100. Although the MEIB objective still performs better than the VIB counterpart with the HIB framework in the case of D=100, we realize that the argument was rather strong as a general statement and as we did not cover all possible cases. Therefore, we have removed the statement. Thank you for commenting on this important point.**
>
> - In Fig. 9, should not at α = 0 the performance of VIB and MEIB be identical? Can you explain the difference?
>
> **We agree that their performance should be identical at α = 0 if the identical architecture was used. The difference came from the fact that we used different architectures, with/without a Batch Normalization layer, by default, as we described in the hyperparameter study**
>
> - Why is the method called "maximum entropy information bottleneck" and not just "maximum entropy bottleneck"? After all, instead of (mutual) information, now the entropy is used for regularization.
>
> **Although entropy is the main regularization term used, we think MEIB succeeds and improves upon IB and VIB. Thus, we call it as “Maximum Entropy” regularized “Information Bottleneck” family.**
>
> - Page 4: σ2I is misleading, as it suggests that all elements on the main diagonal have the same variance.
>
> **We have changed the notation as Diag(\sigma_\theta^2).**
>
> - Reference Thomas & Joy (2006) is broken.
>
> **We have fixed it. Thank you so much for the detailed review.**

---

> > ### Comment · Reviewer_PNBE · 2022-11-21
> > **Thanks**
> >
> > I thank the authors for the answers to some of my concerns. Since some of my more critical concerns have not been addressed in the rebuttal, I will keep my score as is.

---

### Official Review · Reviewer_YBtS · 2022-10-20

**Confidence:** 5
**Correctness:** 4
**Technical Novelty And Significance:** 3
**Empirical Novelty And Significance:** 3
**Recommendation:** 8

**Clarity, Quality, Novelty And Reproducibility:**

The paper is very clearly written and of high quality in all respects. It is novel enough and totally reproducible.

**Strength And Weaknesses:**

I liked very much the idea, both the simplicity and the motivation behind it. I also found very intriguing the property of MEIB being a lower bound on VIB. The experiments are extensive, combined with deep ablation studies; they convincingly show that models trained with the MEIB objective outperform existing methods in terms of regularization, perturbation robustness, probabilistic contrastive learning, and risk-controlled recognition performance.


**Summary Of The Paper:**

The paper presents a Maximum Entropy Information Bottleneck (MEIB), which is a different take on the information bottleneck compared to the well-known variational information bottleneck.




**Summary Of The Review:**

A high quality paper that may spur new interest in information bottleneck techniques for deep networks.

---

> ### Author Response · Authors · 2022-11-19
> **Response to Review**
>
> It is our pleasure to hear that you enjoyed our research and paper.
> We really appreciate your time in reading and reviewing our paper.
> If you have any questions, please let us know.

---

### Official Review · Reviewer_yBZK · 2022-10-25

**Confidence:** 4
**Correctness:** 3
**Technical Novelty And Significance:** 3
**Empirical Novelty And Significance:** 3
**Recommendation:** 5

**Clarity, Quality, Novelty And Reproducibility:**

The manuscript is easy to follow. The method is also easy to be reimplemented, although authors did not attach the code.

**Strength And Weaknesses:**

Strength:
1. A simple method but works well. It is also good that authors evaluated on a real ReID task.
2. The manuscript is easy to follow.

Weaknesses:
1. In fact, it is easy to verify that the basic IB objective is also an upper bound of the so-called MEIB:
$L_{IB}=I(Z,Y)-\beta* I(Z,X)=I(Z,Y)-\beta* [H(Z)-H(Z|X)]=[I(Z,Y) + \beta* H(Z|X)] - \beta* H(Z)= L_{MEIB}-\beta* H(Z)>=L_{MEIB}$
The difference is just the absence of negative entropy of $Z$.

Authors have empirically demonstrated the effectiveness of the MEIB and its better performance of robustness as well as accuracy in specified scenarios. It would be more convincing if the authors present some theoretical explanations on why MEIB performs better than IB or VIB by just removing the regularization of H(Z) or the cross-entropy term.

I can expect MEIB provide more accurate estimation on input uncertainty. However, it is hard for me to understand why MEIB is also robust to adversarial pertubations. Figure 3 is confusing to me, especially the VIB performs worst in adversarial robustness. This observation actually conflicts with previous literature (such as nonlinear IB or HSIC-bottleneck or even the classic VIB) which suggests that one can improve adversarial robustness by a regularization on H(X;Z) or even H(Z).

2. The name (i.e., maximum entropy information bottleneck) of new method is also a bit confusing to me. I am not sure if the new method can be interpreted as a bottleneck or not. Usually, the IB or VIB involves a trade-off by maximizing a term, while also constraining another term. However, the new objective actually maximizes both I(Z,Y) and H(Z|X), i.e., there seems no bottleneck effect.

3. In Section 4.2, can you validate that "MEIB or its variant should perform better with much higher dimensional embeddings"?


**Summary Of The Paper:**

This manuscript presents a new stochastic embedding method named Maximum Entropy Information Bottleneck (MEIB). Authors show that the well-known VIB is an upper bound of MEIB. Experimental results show that MEIB performs more robustly against adversarial attacks or other types of perturbation than VIB. The proposed method also outperforms the other four algorithms in the ReID missions on all three datasets.

**Summary Of The Review:**

A simple but effective method for stochastic embedding. Some results are not well justified.

---

> ### Author Response · Authors · 2022-11-19
> **Response to Review**
>
> **Thank you for your time in reviewing our manuscript and your valuable comments. Please find answers to some of the questions posed.**
>
> - In fact, it is easy to verify that the basic IB objective is also an upper bound of the so-called MEIB: The difference is just the absence of negative entropy of Z.
>
> **We agree that the derivation given by the reviewer is valid if we assume discrete random variables. For continuous random variables, however, the (differential) entropy term, H(Z), can be either negative or positive (Thomas & Joy, 2006). In that case, β∗H(Z) cannot be simply eliminated from the inequality. Thus, we derived the bound of MEIB in a bit complicated way as described in the manuscript.**
>
> - Authors have empirically demonstrated the effectiveness of the MEIB and its better performance of robustness as well as accuracy in specified scenarios. It would be more convincing if the authors present some theoretical explanations on why MEIB performs better than IB or VIB by just removing the regularization of H(Z) or the cross-entropy term.
>
> **At the moment, we have been able to theoretically incorporate entropy maximization into the information bottleneck framework. Further, we gave empirical evidence of how maximizing entropy expands the coverage in the embedding space. Explaining this theoretically is our current active research. Intuitively, we think the following is the explanation: By maximizing the entropy (spread) of the stochastic embedding of inputs while ensuring that the classification loss is not hurt, the samples are getting mapped far from the decision boundary. We think that this gives more room/higher probability for samples belonging to the same class but with large intra-class variation to be mapped around the mean than away from it. This explains why MEIB is less prone to adversarial attacks. On the other hand, the amount of possible movement (entropy) of input provides a measure of its ‘unambiguity’, which results in better risk-controlled classification performance.**
>
> - I can expect MEIB provide more accurate estimation on input uncertainty. However, it is hard for me to understand why MEIB is also robust to adversarial pertubations.
>
> **Adversarial perturbations, especially the one-step attack like FGSM, modifies the input by small values. These small changes in the input space also move their location in the embedding space by certain amount. If we train a model deterministically or with only a little stochasticity, there could be a higher chance that this movement crosses the decision boundary. On the other hand, if we train a model with maximum possible stochasticity in the embedding space (MEIB) as long as it does not hurt much the task performance, we will have more elbowroom from the mean in the embedding space. Therefore, the chance of MEIB crossing the decision boundary would be less than the deterministic models and the models with small amount of stochasticity. This intuition is depicted in Figure 2.**
>
> - The name (i.e., maximum entropy information bottleneck) of new method is also a bit confusing to me. I am not sure if the new method can be interpreted as a bottleneck or not. Usually, the IB or VIB involves a trade-off by maximizing a term, while also constraining another term. However, the new objective actually maximizes both I(Z,Y) and H(Z|X), i.e., there seems no bottleneck effect.
>
> **We recognize that it could be confusing because both terms are maximized together. However, we consider that there is still a bottleneck effect; to learn an easier decision boundary while maximizing I(Z, Y), it is more convenient for the model to have a small H(Z|X). In other words, larger H(Z|X) means more uncertainty in the embedding, hence harder predictability. On the other hand, we cannot infinitely increase H(Z|X) without compromising on I(Z,Y) maximization. Therefore, there is a trade-off, or a bottleneck, between these two maximization terms.**
>
> - In Section 4.2, can you validate that "MEIB or its variant should perform better with much higher dimensional embeddings"?
>
> **To validate the case with a higher dimension, we have added the result with D=100. Although the MEIB objective still performs better than the VIB counterpart with the HIB framework in the case of D=100, we realize that the argument was rather strong as a general statement and as we did not cover all possible cases. Therefore, we have removed the statement. Thank you for commenting on this important point.**

---

### Official Review · Reviewer_HaaB · 2022-10-31

**Confidence:** 3
**Correctness:** 3
**Technical Novelty And Significance:** 2
**Empirical Novelty And Significance:** 2
**Recommendation:** 3

**Clarity, Quality, Novelty And Reproducibility:**

*Some clarification questions*
- The motivation of using maximum entropy based approach can be described more clearly. Though the results showed better robustness results compared to VIB, it is unclear to me that what leads authors to use maximum entropy
- In Figure 3(b), it is interesting to see when rejection rate > 90%, MEIB yields a larger error rate than other approaches which contradicts the  trend when rejection rate <70% (when > 70, MEIB already performs worse than dropout), any educated guess?
- From figure 4, MEIB do cover larger area in embedding space than VIB. What if applying L2_NORM after last FC before logits layer. Then I assume both approaches perform similar? How to justify the effectiveness in this case? typically people use normalized embedding for retrieval as one use case.
- Given MEIB is essentially increase \delta in VIB adaptively, out of curiosity, whether a periodically update of \delta based on VIB can perform similar, I believe there might be work investigating this.

**Strength And Weaknesses:**

*Strength*
- It is good to see the relationship between MEIB and VIB mathematically to understand the proposed approach better.
- MEIB significantly improve the robustness of the embeddings compared to VIB, dropout and deterministic
- MEIB performs much better than peer approaches in person re-id datasets

*Weaknesses*
- The proposed approach is incrementally novel from VIB. Please clarify the highlight of the motivation. Even mathematically, MEIB and VIB look similar, the motivation matters to justify the novelty.
- Classification performance wise, MEIB is on-par with VIB in MNIST, though the robustness is much better, the trade-off is not discussed. Instead it seems a binary decision instead of a dynamic trade-off between error rate and robustness by tuning any hyperparameter. A suggestion is It might be nice to re-consider the equation (8) to add a (\lambda * prior) item for an unmature example to mitigate this gap
- More to see clarity section

**Summary Of The Paper:**

Stochastic embeddings are explored to benefit its capability of associating uncertainty for robustness to noisy data. Instead of using a standard normal distribution prior for variational information bottleneck principle (that typically use KL to learn), this work uses maximum entropy as bottleneck objective to learn the embedding. The proposed approach outperforms prior based approach in terms of robustness.

**Summary Of The Review:**

This work is self-contained and well written with sufficient discussion to support the claim, yet the novelty is limited and motivation is not very clear. Before the questions are answered, I would suggest a borderline reject.

After discussion and reviewing authors feedback, the feedback answers most of my questions, but did not address the weakness part especially did not convince me of the sufficient novelty. Thus I'll clear update my recommendation as reject. This is a good paper, but not good enough.

---

> ### Author Response · Authors · 2022-11-19
> **Response to Review**
>
> **Thank you for your time in reviewing our manuscript and your valuable comments. Please find answers to some of the questions posed.**
>
> - The motivation of using maximum entropy based approach can be described more clearly. Though the results showed better robustness results compared to VIB, it is unclear to me that what leads authors to use maximum entropy
>
> **The main intuition of MEIB is to spread out each input embedding as wide as possible to take more space within a decision region. By maximizing the conditional entropy of the stochastic embeddings, we would have a better regularization effect as it makes the area secured by the embedding distribution for the given input as broad as possible. For example, adversarial perturbations, especially the one-step attack like FGSM, modifies the input by small values. These small changes in the input space also move their location in the embedding space by certain amount. If we train a model deterministically or with only a little stochasticity, there could be a higher chance that this movement crosses the decision boundary. On the other hand, if we train a model with maximum possible stochasticity in the embedding space (MEIB) as long as it does not hurt much the task performance, we will have more elbowroom from the mean in the embedding space. Therefore, the chance of MEIB crossing the decision boundary would be less than the deterministic models and the models with small amount of stochasticity. This intuition is depicted in Figure 2.**
>
> - In Figure 3(b), it is interesting to see when rejection rate > 90%, MEIB yields a larger error rate than other approaches which contradicts the trend when rejection rate <70% (when > 70, MEIB already performs worse than dropout), any educated guess?
>
> **We have attached some examples of misclassified inputs with a rejection rate of 80% in Section I of the appendix. The shape of the digit in each of those images is very unusual and even looks like the predicted digit. Thus, we guess the MEIB model was overconfident about these images with the predicted classes instead of regarding them as uncertain images. It would be advisable to avoid this behavior also, which we will further investigate in future work.**
>
> - From figure 4, MEIB do cover larger area in embedding space than VIB. What if applying L2_NORM after last FC before logits layer. Then I assume both approaches perform similar? How to justify the effectiveness in this case? typically people use normalized embedding for retrieval as one use case.
>
> **In Section J of the appendix, we have added a set of experimental results by applying L2 normalization to MEIB and VIB. While MEIB still outperforms VIB in this setting, it seems MEIB performs worse with L2 normalization than without it. VIB also could not benefit much from the normalization. We think it is difficult for stochastic embeddings, especially MEIB, to utilize their advantage in angular space since their magnitudes are not considered anymore. For example, the main intuition of MEIB is to spread out each input embedding as wide as possible to take more space within a decision region. However, after applying L2 normalization, only the angle of each feature vector contributes to the decision regardless of their magnitude in the embedding space.**

---

### Decision · Program_Chairs · 2023-01-20

**Decision:**

Reject

**Justification For Why Not Higher Score:**

The reviewers note that the novelty is limited, and that motivation for the method is not very clear.

**Justification For Why Not Lower Score:**

N/A

**Metareview: Summary, Strengths And Weaknesses:**

Ratings: 3/5/8/5
Confidences: 3/4/5/4
Recommendation: Reject

This paper proposes replacing the Variational Information Bottleneck (VIB) with an Maximum Entropy Information Bottleneck (MEIB). The paper shows that the approach results in a higher adversarial robustness and risk-controlled recognition performance.

The reviewers write that the work is well-written and an interesting adaption of VIB, with sufficient discussion to support the claim. However, the reviewers note that the novelty is limited, and that motivation for the method is not very clear.  The authors provided a rebuttal to all reviews, but not further written discussion followed. After the AC/Reviewer meeting, one reviewer reduced his rating to a reject rating.

**Summary Of Ac-Reviewer Meeting:**

An AC/Reviewer discussion was planned, but the AC was too sick to attend. Two reviewers attended. The summary of the meeting: "I met with [other reviewer]. After reviewing the feedback, the critical concerns remain. We both agree that the paper is good, but may not be good enough (novelty is one of the critical concerns), which echoed the original rating for "marginal below acceptance". I changed my rating to reject to make it clear for AC to make the final decision."